# Reductions of Circulating Nitric Oxide are Followed by Hypertension during Pregnancy and Increased Activity of Matrix Metalloproteinases-2 and -9 in Rats

**DOI:** 10.3390/cells8111402

**Published:** 2019-11-07

**Authors:** Regina A. Nascimento, Jose S. Possomato-Vieira, Giselle F. Bonacio, Elen Rizzi, Carlos A. Dias-Junior

**Affiliations:** 1Department of Pharmacology, Biosciences Institute of Botucatu, Sao Paulo State University – UNESP, Botucatu, Sao Paulo 18.618-689, Brazil; apnascimentofarma@hotmail.com (R.A.N.); sergiopossomato@yahoo.com.br (J.S.P.-V.); 2Unit of Biotechnology, University of Ribeirao Preto, UNAERP, Ribeirao Preto, Sao Paulo 14096-900, Brazil; gifbonacio@gmail.com (G.F.B.); ersanchez@unaerp.br (E.R.)

**Keywords:** hypertensive pregnancy, metalloproteinases, nitric oxide, rats

## Abstract

Hypertensive pregnancy has been associated with reduced nitric oxide (NO), bioavailability, and increased activity of matrix metalloproteinases (MMPs). However, it is unclear if MMPs activation is regulated by NO during pregnancy. To this end, we examined activity of MMP-2 and MMP-9 in plasma, placenta, uterus and aorta, NO bioavailability, oxidative stress, systolic blood pressure (SBP), and fetal-placental development at the early, middle, and late pregnancy stages in normotensive and Nω-Nitro-L-arginine methyl-ester (L-NAME)-induced hypertensive pregnancy in rats. Reduced MMP-2 activity in uterus, placenta, and aorta and reduced MMP-9 activity in plasma and placenta with concomitant increased NO levels were found in normotensive pregnant rats. By contrast, increased MMP-2 activity in uterus, placenta, and aorta, and increased MMP-9 activity in plasma and placenta with concomitant reduced NO levels were observed in hypertensive pregnant rats. Also, elevated oxidative stress was displayed by hypertensive pregnant rats at the middle and late stages. These findings in the L-NAME-treated pregnant rats were also followed by increases in SBP and associated with fetal growth restrictions at the middle and late pregnancy stages. We concluded that NO bioavailability may regulate MMPs activation during normal and hypertensive pregnancy.

## 1. Introduction

During pregnancy, the uterus undergoes significant and expansive hypertrophy and distension to provide the adequate space for the growing fetus, while placental remodeling and cytotrophoblast invasion of spiral arteries occur to supply blood flow required for the fetal developing [1,2]. Also, hemodynamic changes as increases in maternal blood volume and cardiac output are counterbalanced by systemic vasodilation following decreases in vascular resistance occur during the gestational period [3,4]. These utero-placental and vascular changes may be modulated by nitric oxide (NO) [5,6] and matrix metalloproteinases (MMPs) [7]. In particular, the gelatinases MMP-2 and MMP-9 have been implicated in these pregnancy-associated changes [8,9]. However, it is unclear whether the activity of MMP-2 and MMP-9 is regulated by endogenous NO during pregnancy.

There are increases in endogenous NO synthesis in normotensive pregnancy [5,6,10,11], while hypertensive disorders of pregnancy are associated with reductions of NO bioavailability in humans [12,13,14,15,16] and in animals [17,18,19]. Since healthy gestations undergo significant utero-placental and vascular adaptations [7,20] and that MMPs may also play important roles in tissue remodeling [21], one may consider that changes in activity of MMPs may occur during normal and hypertensive pregnancy [7,20].

In normal pregnancy, previous studies have found that early-pregnant animals present increases in the activity of MMP-2 in uterus [22] and increases in placental MMP-9 [23]. Also, other authors reported increases in activity of MMP-2 and MMP-9 in the uterus and aorta during middle and late pregnancy stages, but decreases in placental activity of these two gelatinases were observed in late compared to middle pregnancy stages in animals [24,25]. Furthermore, in hypertensive disorders of pregnancy, excessive gelatinolytic activity of MMP-2 and MMP-9 in plasma [26,27], and increases in MMP-2 in urine [28] have been found in women. Of note, these earlier studies have focused on the end of pregnancy, however, considering the potential role of NO to downregulate the MMP-2 and MMP-9 [29,30,31,32,33,34], we hypothesized that reductions in NO bioavailability caused by Nω-Nitro-L-arginine methyl ester (L-NAME) [35,36] could increase the activity of MMPs throughout pregnancy stages, since L-NAME-induced hypertensive pregnancy causes fetal-placental growth restriction because of decreases in utero-placental perfusion, which is the result of aberrant vascular remodeling in the utero-placental circulation [37,38].

Therefore, this study aimed to examine whether decreases in circulating NO may be associated with changes in activities of MMP-2 and MMP-9 in early, middle, and late stages of hypertension in pregnancy. To this end, we investigated the changes in systolic blood pressure, NO bioavailability and oxidative stress in plasma as well as gelatinolytic activities of MMP-2 and MMP-9 in plasma and in important tissues that undergo extensive remodeling and physiological adaptations during pregnancy, including uterus, placenta, and abdominal aorta in specific stages of normotensive pregnancy and L-NAME-induced hypertensive pregnancy in rats.

## 2. Materials and Methods

### 2.1. Animals

Three-month-old female Wistar rats (200–250 g) were housed in animal facility and maintained in 22 ± 2 °C (12-hour light-dark cycle) and given free access to standard rat chow and tap water. Experiments with virgin rats were conducted during the estrous phase in order to control for reproductive cycle and endocrine confounders, vaginal secretion of female rats was collected daily in the morning, and an estrus smear primarily consisted of anucleated cornified squamous cells. This was confirmed prior to all experimentations [39]. The experiments with pregnant rats were conducted as the following: each female rat was separately mated with a male rat overnight, and the first pregnancy day was defined when spermatozoa and estrous cells were found in a vaginal smear.

### 2.2. Experimental Design

The experimental groups consisted of age-matched virgin and pregnant rats that were randomly divided in 8 groups (n = 8 animals per group) as the following: Normotensive virgin rats (Virgin group) received daily intraperitoneal (i.p.) injections of saline for 6 consecutive days;Hypertensive virgin rats (Virgin+L-NAME group) received daily i.p. injections of L-NAME for 5 consecutive days;Pregnant rats were studied according to the different pregnancy stages, which correspond to the three gestational trimesters:
Normotensive early pregnancy (Early-Preg group): pregnant rats received daily i.p. injections of saline from pregnancy day 3 to 8;Hypertensive early pregnancy (Early-Preg+L-NAME group): pregnant rats received daily i.p. injections of L-NAME from pregnancy day 3 to 8;Normotensive middle pregnancy (Mid-Preg group): pregnant rats received daily i.p. injections of saline from pregnancy day 9 to 15;Hypertensive middle pregnancy (Mid-Preg+L-NAME group): pregnant rats received daily i.p. injections of L-NAME from pregnancy day 9 to 15;Normotensive late pregnancy (Late-Preg group): pregnant rats received daily i.p. injections of saline from pregnancy day 13 to 20;Hypertensive late pregnancy (Late-Preg+L-NAME group): pregnant rats received daily i.p. injections of L-NAME from pregnancy day 13 to 20.

The dose of the L-NAME (60 mg/Kg/daily) was based on previous studies [40]. Moreover, experimental design was based on previous studies [35,36] and we included virgin rats in order to investigate whether L-NAME could have the same capacity to induce hypertension in virgin rats as well as whether hormonal changes during pregnancy stages could significantly affect the effects of L-NAME at the different pregnancy stages.

### 2.3. Blood Pressure Measurements 

Systolic blood pressure was recorded daily in virgin rats from days 1 to 6 and in pregnant rats on gestational days 3, 4, 6 and 8 (early pregnancy stage), 9, 11, 13 and 15 (middle pregnancy stage) and 13, 14, 16, 18 and 20 (late pregnancy stage). Systolic blood pressure was measured by tail cuff plethysmography (Insight, Ribeirao Preto, Sao Paulo, Brazil, # EFF-306) [41,42].

Animals were “trained” using the measurement process for 3 days before the beginning of the treatments (data were discarded). The first measure of systolic blood pressure was recorded in all animals, prior to randomization into saline-treated or L-NAME-treated groups, before i.p. injections in order to represent the baseline values (BL) on day 1 (Virgin and Virgin+L-NAME groups), on day 3 (Early-Preg and Early-Preg+L-NAME groups), on day 9 (Mid-Preg and Mid-Preg+L-NAME groups) and on day 13 (Late-Preg and Late-Preg+L-NAME groups). The other measurements were taken 6 hours after i.p. injections. Briefly, rats were restrained and pre-warmed in a warm-box (Insight, Ribeirao Preto, Sao Paulo, Brazil, #EFF-307) at 40 °C for 10 min. Then, systolic blood pressure was determined as the average of the cuff inflation-deflation (3–6 cycles) by a trained operator. 

### 2.4. Animal Euthanasia and Tissue Collection

On the last day of each experimental group, dams were killed under an overdose of isoflurane, followed by exsanguination. Blood samples were collected in lyophilized ethylenediaminetetraacetic acid (EDTA) (Vacuntainer Becton-Dickinson, BD, Oxford, UK) and lyophilized heparin containing tubes (Vacuntainer Becton-Dickinson, BD, Oxford, UK) and immediately centrifuged. Plasma was separated and stored at −80 °C until use for biochemical analysis. 

The abdominal cavity was opened, and the abdominal aorta, uterus and placentae were rapidly excised and placed in cold Krebs solution. With the aid of a dissection microscope, the virgin uterus was cut into 5 mm wide strips. The pregnant uterus was cut and opened, and the placentae and pups were removed. The uterus was then segmented along its longitudinal axis into 5 mm long and 5 mm wide strips, and we separated the circular muscle layer from longitudinal muscle and removed the endometrium lining from the uterine strip. The placenta was cut into 5 mm × 5 mm strips. The abdominal aorta was cleaned of connective and perivascular adipose tissue and cut into 5 mm rings. Plasma samples and 8 to 12 uterine, placental and abdominal aorta segments from each pregnant rat as well as plasma samples, uterine and abdominal aorta segments from each virgin rat were stored at −80 °C for biochemical analysis.

### 2.5. Fetal and Placental Parameters

Litter size (number of pups) was observed in all stages of pregnancy, while a number of viable fetuses, number of resorptions, and fetal and placental weights were observed only on the middle and late pregnancy stages. Viable fetuses were determined as those which showed no macroscopic signal of malformation and could apparently have a normal outcome with the advancement of the pregnancy, as previously reported [43].

### 2.6. Determination of Plasma Levels of Nitrite/Nitrate (Total NOx)

Plasma total NOx concentrations were determined using Griess reagents followed by reduction of nitrous species with vanadium chloride III [44]. Briefly, before the addition of Griess reagents, samples were incubated with 100 µL of saturated solution of vanadium chloride III for three hours at 37 °C with agitation. After incubation, 50 µL of 1% sulfanilamide solution in 5% phosphoric acid was added and microplate was incubated for 10 minutes protected from light. Then, 50 µL of 0.1% N-(1-Naphthyl)-ethylenediaminedihydrochloride solution were added followed by 10-minute incubation in the dark. Absorbance at 535 nm was read in spectrophotometer (Synergy 4, BIOTEK, Winooski, VT, USA) and NOx concentration was calculated using a standard curve of sodium nitrite (1.56–100 µM). The metabolites levels of NO (NOx) in plasma were expressed in μmol/L.

### 2.7. Determination of Lipid Peroxidation

Lipid peroxidation was assessed through measurements of thiobarbituric acid reactive substances (TBARS) [45]. Thiobarbituric acid (TBA) reacts with malondialdehyde (MDA), which is the end product of lipid peroxidation to form a magenta color that is read by spectrophotometer at 532 nm (Synergy 4, BIOTEK, Winooski, VT, USA). In test tubes, a reaction mixture containing 100 µL of distilled water, 50 µL of 8.1% sodium dodecyl sulfate (SDS), 100 µL of plasma samples, 375 µL of acetic acid 20% and 375 µL of TBA 0.8% were incubated in water-bath at 95 °C for one hour and thereafter, mixture was centrifuged at 4,000 rpm for 10 minutes. Standard curve was made in a similar form, replacing samples with 25 µL of known concentrations of MDA. Plasma levels of TBARS were calculated compared to a standard curve of MDA (20–320 nmol). TBARS concentrations were expressed in nmol/mL.

### 2.8. Zymography for MMP-2 and MMP-9 Activity

Gelatin zymography was used to determine MMP-2 and MMP-9 activity in plasma, placenta, uterus and aorta as previously described [46,47,48,49]. Briefly, frozen samples were homogenized in ice-cold RIPA buffer (1 mM 1,10-ortho-phenanthroline, 1mM phenylmethanesulfonyl fluoride, and 1mM N-ethylmaleimide; Sigma-Aldrich, St. Louis, MO, USA) and protease inhibitor (4-(2-aminoethyl) benzenesulfonyl fluoride (AEBSF), E-64, bestatin, leupeptin, aprotinin, and EDTA) in a proportion of 100 μL RIPA + protease inhibitor for each 10 mg of tissue sample. The samples were placed in ice for 2 hours with gentle stirring and then centrifuged at 12,000 rpm for 10 minutes. The protein concentrations were measured using the Bradford assay (Sigma-Aldrich). 10 µg of uterus and aorta proteins and 5 µg of placenta proteins were diluted 1:1 with sample buffer (final concentration): 2% SDS (sodium dodecyl sulfate), 125 mM Tris–HCl, pH 6.8, 10% glycerol, and 0.001% bromophenol blue, and were subjected to electrophoresis on 12% SDS–polyacrylamide gel electrophoresis copolymerized with gelatin (0.05%) as the substrate. After electrophoresis was completed, the gels were incubated twice for 30 min at room temperature in a 2% Triton X-100 solution, washed, and incubated at 37 °C for 18 hours in Tris–HCl buffer, pH 7.4, containing 10 mmol/L CaCl_2_. Gels were stained with 0.05% Coomassie Brilliant Blue G-250 and then destained with 30% methanol and 10% acetic acid. Gelatinolytic activity was detected as an unstained band against the background of Coomassie blue-stained gelatin. Enzyme activity was assayed using ImageJ software by optical densitometry and the integrated protease activity density was measured as pixel intensity × mm^2^. Gelatinolytic activities were normalized with regards to an internal standard (2% fetal bovine serum, FBS) to correct for loading and inter-gel variation, and the results were expressed as arbitrary units. The MMP-2 forms were identified as bands at 75, 72 and 64 kDa and MMP-9 as band at 92 KDa. Total MMP-2 was taken as a sum of the different isoforms.

### 2.9. Statistical Analysis

Using commercially available statistical software (Graph Pad Prism® 6.0 for Windows, San Diego, CA, USA), a Shapiro-Wilk test was applied to verify normality of data distribution. Comparisons among groups were assessed by two-way analysis of variance (ANOVA) or one-way ANOVA. When a statistical difference was observed, data were further analyzed using Tukey’s correction for multiple comparisons. Student’s t-test was used for comparison of two means. A probability value *P* < 0.05 was considered statistically significant. All values are expressed as mean ± S.E.M.

## 3. Results

### 3.1. L-NAME Treatment Presented Hypertnsion in Virgin and Early, Middle and Lat Pregnancy Stages

In virgin rats, increases in systolic blood pressure occurred after one day of L-NAME injections when compared to day one in Virgin group (**P* < 0.05), and systolic blood pressure was maintained elevated in Virgin+L-NAME compared to respective days in Virgin group (*^,#^*P* < 0.05, Figure 1A). 

In the early pregnancy stage, increases in systolic blood pressure were observed only after five days of L-NAME injections on gestational day 8 compared to gestational day 3 (**P* < 0.05) or the respective day in the Early-Preg group (^#^*P* < 0.05, Figure 1A).

In middle pregnancy stage, systolic blood pressure increased after four days of L-NAME injections on gestational day 13 and was maintained elevated on gestational day 15 compared to gestational day 9 (**P* < 0.05) or respective days in Mid-Preg group (*^,#^*P* < 0.05, Figure 1A).

In late pregnancy stage, increases in systolic blood pressure occurred after one day of L-NAME injection on gestational day 14 compared to gestational day 13 (**P* < 0.05) and were maintained at an elevated level in Late+L-NAME compared to respective days in the Late-Preg group (*^,#^*P* < 0.05, Figure 1A). 

### 3.2. Circulating NO Increases in Middle and Late but Not in Early Pregnancy Stage, While Decreases in Circulating NO Are Observed in Virgin Rats and in Middle and Late Pregnant Rtas Treated with L-NAME. But, Early Pregnant Rats Treated (Or Not) with L-Name Presented Similar Circulating NO

The NO bioavailability showed significant increases in Late-Preg compared to Virgin, Mid-Preg, and Early-Preg groups (*^, +, ++^*P* < 0.05, Figure 1B) and in Mid-Preg compared to Early-Preg group (^+^*P* < 0.05, Figure 1B). Moreover, reduced NO levels were found in Virgin+L-NAME, Mid-Preg+L-NAME, Late-Preg+L-NAME but not in Early-Preg+L-NAME compared to the respective saline-treated group (*^, #^
*P* < 0.05, Figure 1B).

### 3.3. Intrauterine Growth Restrictions Are Found in Middle and Late Pregnant Rats Treated with L-NAME

Fetal but not placental parameters were negatively affected by L-NAME, presenting significant reductions in litter size (^#^*P* < 0.05, Figure 2A) and number of viable fetuses (^#^*P* < 0.05, Figure 2B) with concomitant increases in number of resorptions (^#^*P* < 0.05, Figure 2C). Also, fetal weight (^#^*P* < 0.05, Figure 2D) but not placental weight (Figure 2E) presented reductions in Mid-Preg+L-NAME and Late-Preg+L-NAME compared to Mid-Preg and Late-Preg groups (^#^*P* < 0.05, Figure 2D).

### 3.4. Oxidative Stress Is Increased in Middle and Late Pregnant Rats Treated with L-NAME as Well as in Late Pregnant Rats Treated with Saline. However, Oxidative Stress Is Reduced in L-NAME-Treated Virgin Rats, While No Significant Difference Is Found in Early Pregnancy Stage Treated (Or Not) with L-NAME

In normal pregnant rats, oxidative stress was reduced in Early-Preg and Mid-Preg groups, while increases in Late-Preg were found compared to the Virgin group (**P* < 0.05, Figure 3). Moreover, oxidative stress in Late-Preg+L-NAME was also increased compared to Early-Preg+L-NAME and Mid-Preg+L-NAME groups (^+, ++^*P* < 0.05, Figure 3). Additionally, Virgin+L-NAME and Early-Preg+L-NAME presented reduced oxidative stress compared to the Virgin group (**P* < 0.05, Figure 3). Furthermore, Mid-Preg+L-NAME presented increases in oxidative stress compared to the Mid-Preg group (^#^*P* < 0.05, Figure 3).

### 3.5. Changes in Activity of MMP-9 in Plasma Are Found Only at the Late Pregnancy Stage, in Which a Decrease Is Observed in Saline-Treated Pregnant Rats, While Increases Are Observed in L-NAME-Treated Pregnant Rats Compared to the Virgin Group. However, No Differences Are Found in the Activity of MMP-2 in Plasma

A representative zymogram of gelatinolytic activity of plasma samples is shown in Figure 4A. Corresponding bands to ~135 kDa, 92 KDa pro-MMP-9 and three isoforms of MMP-2: 75 KDa, 72 KDa and 64 KDa were observed in FBS, so it was used as a positive control to normalize the inter-gel activities of these MMPs. No corresponding band to ~135 kDa and 64 KDa MMP-2 appeared in zymography gels. Saline-treated pregnant rats presented reductions in MMP-9 activity only in Late-Preg compared to Virgin group (**P* < 0.05, Figure 4B). However, an increase in MMP-9 activity was found only in Late-Preg+L-NAME compared to the Late-Preg group (^#^*P* < 0.05, Figure 4B). No significant differences were observed at the early and mid-pregnancy stages (*P* > 0.05, Figure 4B–F).

### 3.6. Uterine MMP-9 Activity Is Only Detected in L-NAME-Treated Pregnant Rats, While Uterine MMP-2 Is Reduced in Middle and Late, but Not in Early Pregnant Rats Treated with Saline. However, L-NAME-Treated Pregnant Rats Presented Increases in MMP-2 Activity in Early, Middle, and Late Pregnancy Stages

A representative zymogram of gelatinolytic activity of uterine tissue homogenate is shown in Figure 5A. Gels analysis showed no corresponding band to 92 KDa pro-MMP-9 in all groups, except in Late-Preg+L-NAME group (Figure 5A), thus, zymography analysis shows the isoform of 92 kDa MMP-9 only in the Late-Preg+L-NAME group (Figure 5B), and three isoforms of MMP-2: 75, 72, and 64 kDa. Saline-treated pregnant rats presented reductions in activity of all MMP-2 isoforms and total MMP-2 in Mid-Preg and Late-Preg (**P* < 0.05), but not in Early-Preg compared to the Virgin group (Figure 5C–F). However, increased activity of 75 and 72 KDa MMP-2 were observed in all L-NAME-treated pregnant rats (early-, mid-, and late-pregnancy) compared to the respective saline-treated pregnant group (^#^*P* < 0.05, Figure 5C–D). Also, increased activity of 64 KDa and total MMP-2 were observed in Mid-Preg+L-NAME and Late-Preg+L-NAME, but not in Early-Preg+L-NAME compared to the respective saline-treated pregnant group (^#^*P* < 0.05, Figure 5E,F).

### 3.7. Placental MMP-9 and MMP-2 Activities Are Decreased in Saline-Treated Pregnant Rats, While Increases Are Found in L-NAME-Treated Pregnant Rats

A representative zymogram of gelatinolytic activity of placental tissue homogenate is shown in Figure 6A. Corresponding band to 92 KDa pro-MMP-9 and three isoforms of MMP-2: 75 KDa, 72 KDa, and 64 KDa (Figure 6A). Saline-treated pregnant groups showed decreased activity of MMP-9 (Figure 6B), 64 KDa (Figure 6E) and total MMP-2 (Figure 6F) in Late-Preg compared to the respective Mid-Preg group (^+^*P* < 0.05). Also, increased activity of MMP-9 was only observed in Late-Preg+L-NAME group (Figure 6B) and increased activities of all isoforms and total MMP-2 (Figure 6C–F) were found in Mid-Preg+L-NAME and Late-Preg+L-NAME compared to the respective saline-treated group (^#^*P* < 0.05). Moreover, Late-Preg+L-NAME presented a lower activity of 64 kDa and total MMP-2 compared to the respective Mid-Preg+L-NAME group (^++^*P* < 0.05).

### 3.8. MMP-9 Is Not Detected in the Aorta. Moreover, MMP-2 Activity Is Reduced in Saline-Treated Pregnant Rats, While Increased MMP-2 Activity Is Observed in L-NAME-Treated Pregnant Rats

A representative zymogram of gelatinolytic activity of aorta homogenate is shown in Figure 7A. No corresponding band to 92 KDa pro-MMP-9 appeared in zymography gel, thus zymogram shows three isoforms of MMP-2: 75, 72, and 64 kDa. Saline-treated pregnant rats presented reductions in the activity of 75 KDa MMP-2 in Early-, Mid-, and Late-pregnant rats compared to the Virgin group (**P* < 0.05, Figure 7C). Also, reduced activity of 72 KDa MMP-2 was found in Mid-Preg and Late-Preg but not in Early-Preg compared to the Virgin group (**P* < 0.05, Figure 7D). Moreover, 64 KDa and total MMP-2 activities showed no differences in saline-treated pregnant rats compared to the Virgin group (Figure 7E,F). However, increased activity of 75, 64 KDa and total MMP-2 were found in Early-Preg+L-NAME, Mid-Preg+L-NAME and Late-Preg+L-NAME compared to respective Early, Mid- and Late-Preg groups (^#^*P* < 0.05, Figure 7C,E,F). Furthermore, increased activity of 72 KDa was observed in Mid-Preg+L-NAME and Late-Preg+L-NAME but not in Early-Preg+L-NAME compared to respective saline-treated groups (^#^*P* < 0.05, Figure 7D).

## 4. Discussion

The main findings of this study are: (1) during normal pregnancy, there are increases in NO bioavailability in middle and late pregnancy stages compared to early pregnancy stage and virgin rats, with concomitant reductions in MMP-9 activity in plasma and placenta at the late pregnancy stage, and reductions in 75 kDa, 72 kDa and 64 kDa and total MMP-2 activities in uterus in middle and late pregnancy stages, while decreases in 75 kDa MMP-2 activity in aorta were found in early, middle and late pregnancy stages and decreases in 72 kDa MMP-2 in middle and late pregnancy stages were also observed. However, (2) L-NAME increased systolic blood pressure in virgin and in early, middle and late pregnancy stages with concomitant decreases in NO bioavailability in virgin rats and in middle and late pregnancy stages but not in the early pregnancy stage; (3) L-NAME negatively affected fetal growth in middle and late pregnancy stages, while no significant changes in placental weights were observed with L-NAME treatment; (4) MMP-9 activities were increased in plasma and placenta only in late pregnant rats treated with L-NAME; (5) 75 and 72 kDa MMP-2 activities were increased in the uterus of early, middle, and late pregnancy stages, while 64 kDa MMP-2 activity increased in middle and late pregnant rats treated with L-NAME, and, total uterine MMMP-2 activity increased only in late pregnant rats treated with L-NAME; placental 75, 72, 64 and total MMP-2 activity were greater in middle and late pregnant rats treated with L-NAME; (6) MMP-9 was not detected in the aorta, while increases in 75 kDa and 64 kDa MMP-2 activities were observed in aorta in early, middle, and late pregnancy stages, while 72 kDa and total MMP-2 activities presented increases in middle and late but not in the early pregnancy stages. The present findings support the hypothesis that NO bioavailability may regulate the activity of MMP-2 and MMP-9 during normotensive and hypertensive pregnancy.

The rises in blood pressure promoted by L-NAME throughout early, middle and late pregnancy stages found in the present study are in accordance with previous reports [35,36,50,51]. Although we did not observe decreases in NO bioavailability in virgin and early-pregnant rats treated with L-NAME, our results regarding to the reduced NO bioavailability followed by increases in systolic blood pressure during middle and late pregnancy stages indicate that the demand for constitutive NOS isoforms-synthesized NO, i.e., by the endothelial (eNOS) and neuronal (nNOS), in the nanomolar range, may be increased at the later stages of pregnancy. Of note, although L-NAME affects all the NOS isoforms (eNOS, nNOS and inducible NOS: iNOS), one could discern that iNOS-derived high NO concentrations, in the micromolar range, possibly could also be demanded during the late stages of a healthy pregnancy. However, a previous study examined the effects of a highly selective iNOS inhibitor with irreversible effects, and was at least 1000-fold more selective for iNOS than for eNOS in healthy pregnant rats for 5 consecutive days [52]. Then, the authors observed no changes in blood pressure, iNOS expression, and oxidative stress in the iNOS inhibitor-treated pregnant rats compared to saline-treated pregnant rats [52]. Also, previous findings showed no significant changes in protein expression of the iNOS and nNOS isoforms during early, middle, and late pregnancy stages in rats [10], as well as in a clinical trial showing that no difference was observed in iNOS expression in normotensive pregnancy volunteers [15].

Our results in normal pregnant rats are supported by previous findings showing increases in NO bioavailability throughout pregnancy stages in rats, exhibiting that eNOS-derived NO was increased at the middle and late pregnancy stages [10], reaching peak levels in middle and late pregnancy stages. Also, NO bioavailability showed similar lower values in virgin and post-partum rats compared to middle and late pregnant rats [10]. Accordingly, increased blood pressure was observed in pregnant eNOS knockout mice, whereas blood pressure remains unaltered in eNOS knockout virgin mice [53]. Importantly, the rises in blood pressure occurred in a more “abrupt” manner in late pregnant rats compared to the latency in increasing the blood pressure observed in virgin rats, early, and middle pregnant rats treated with L-NAME in the present study, further confirming that the demand for eNOS-derived NO may be increased in later pregnancy stages. Furthermore, a clinical study revealed that L-arginine (substrate for eNOS synthesis) was greater in normotensive pregnancy compared to women with preeclampsia, and, also, lower NO levels were observed women with preeclampsia versus normotensive pregnancy [15]. Also, we found that fetal development was negatively affected during middle and late pregnancy stages, whereas the placental growth presented no changes in pregnant rats treated with L-NAME, which confirms a greater involvement of NO at the middle and late pregnancy.

A previous and elegant study accurately examined the placental and fetal oxygenation during pregnancy in mice. Authors combined laser technology with ultrasound in the real time analysis of placental and fetal oxygen saturation in mice treated with L-NAME from pregnancy day 11 to 18 [38]. Then, authors found that L-NAME decreased placental and fetal oxygen saturation and these critical conditions were associated with fetal growth restriction and hypertensive pregnancy [38]. Also, hypoxia-inducible factor 1-α immunostaining was higher in the L-NAME-pregnant group compared to saline-treated pregnant rats [38]. Therefore, we may conclude that L-NAME-induced placental ischemia could have released vasopressors into the maternal circulation that modified the endothelial function, thus altering vasodilators/vasoconstrictors balance [54,55,56,57]. The endothelial dysfunction caused by L-NAME could have affected multiple maternal organs, and the impaired control of vascular function that contributed to hypertension in pregnancy.

Although it is still unclear in the context of hypertension in pregnancy and preeclampsia [28], it is possible that upregulated activity of MMPs also contribute to the increased concentrations of vasoconstrictors and reduced concentrations of vasodilators [58,59,60,61]. Previous studies suggest that MMPs activity imbalance apparently generates vasoconstrictors and degrades vasodilators that may promote vasoconstriction and cause endothelial dysfunction, including in the preeclampsia [62]. While normal trophoblastic invasion of maternal vessels results in extracellular matrix remodeling, which increases the utero-placental vessel distensibility to accommodate the increased blood flow in healthy gestation [63], in preeclampsia, trophoblastic invasion is reduced, leading to incomplete modification of maternal spiral arteries that results in decreases of the placental perfusion [64,65,66]. Moreover, few clinical studies have evaluated the role of MMP-2 and MMP-9 in the pathophysiology of preeclampsia. In this regard, it was demonstrated increased MMP-2 levels in urine of preeclamptic women [28] and increases MMP-2 levels in aminiotic fluid of women who subsequently develop preeclampsia [67]. The same authors examined pro-MMP-9 levels in normal amniotic fluid, but zymogram wells loaded with preeclamptic amniotic fluid did not present any MMP-9 bands [67]. Also, enzyme-linked immunosorbent (ELISA) assays revealed that plasma MMP-9 concentrations were increased in preeclampsia [68]. Interestingly, other clinical trial showed that plasma MMP-9 concentrations are increased in women prior to presentation of preeclampsia [69]. Furthermore, findings at the 36th gestational week MMP-2 concentrations were elevated in preeclampsia [70]. Additionally, increases in MMP-2, but no significant differences in TIMP-2, in preeclamptic patients were observed compared with healthy pregnant women [20]. Altogether, these prior studies suggested that increased activity of MMPs may play roles in healthy cardiovascular adaptation seen in normal pregnancy. However, it is important to note that these elegant previous investigations provided no data related to the NO bioavailability.

There are evidences in vitro showing the involvement of cyclic-GMP-dependent protein kinase (PKG), an important mediator of NO and cGMP signaling pathway in the vascular smooth muscle cells (VSMC), possibly regulates the suppression of MMP-2 through the increases in tissue inhibitors of metalloproteinase (TIMPs), particularly TIMP-2 in rat aortas [71]. In addition, previous studies also showed that NO interferes with nuclear factor Kappa B (NFκB) activity [72], and this transcriptional factor is an important regulator of MMPs expression [73,74,75]. However, other earlier study showed that NO, in a concentration-dependent manner, downregulated MMP-9 activity with concomitant increases in TIMP-1 concentrations in human umbilical vein endothelial cells. Despite this, authors demonstrated that these effects were independent of mechanisms involving cGMP or NFκB [29].

In order to expand the knowledge of MMPs activity regulation in vivo and during pregnancy compared to non-pregnancy conditions, we examined whether NO bioavailability could have an association with changes in the activity of MMPs during normotensive and hypertensive pregnancy compared to virgin rats. Then, gelatinolytic activities of MMP-2 and MMP-9 were examined in target tissues such as the uterus, which undergoes remodeling to accommodate the fetal growing; the placenta, which provides nutrients supply to the fetal developing; and the plasma and aorta, which reflect the vascular changes into maternal circulation. In this regard, we sought to investigate the alterations of the MMP-2 and MMP-9 gelatinolytic activities in specific stages of normal pregnancy and L-NAME-induced hypertensive pregnancy.

In general, we observed that during a healthy pregnancy, there are increases in NO bioavailability followed by decreases in the MMP-2 activity in the uterus, placenta, and aorta and decreases in the MMP-9 activity in plasma and placenta in normotensive pregnant rats. Although some isoforms presented no changes in their activities in normotensive pregnant compared to virgin rats, our present results suggest that the reduced activity of MMPs in healthy pregnant rats may be related to the MMPs regulation mechanisms in middle and late pregnancy stages. It is possible that the aorta, uterus, and placenta as potential sources of MMPs may have finite capacity for gelatinolytic activity and that may be downregulated by the increases in endogenous NO, as demonstrated previously in vitro [29], and our data suggest that downregulation of MMPs activity occurs particularly during the middle and late pregnancy stages. According to this idea, our results also showed no significant changes either in activity of MMPs (in plasma, utero and placenta) or in circulating NO at the early pregnancy stage, implicating a key role played by NO in the downregulation of these gelatinases in the fetal-placental circulation during late pregnancy stages. Also, most of the arterial, uterine and placental remodeling occurs during the peri-implantation period and during fetal and organ development in early and middle gestation and further arterial, uterine and placental remodeling may not be needed during the late pregnancy stage. Interestingly, we unexpectedly also detected decreases in activity of MMP-2 (75KDa) in aorta at the early pregnancy stage compared to Virgin rats, suggesting that this downregulation mechanism may be associated with early pregnancy-induced changes, but it is not related to NO signaling, i.e., during middle and late pregnancy stages endogenous NO synthesis may downregulate MMPs, while during early pregnancy stage this downregulating mechanism may not be dependent of endogenous NO synthesis, but this finding implicates that early pregnancy-induced adaptations protect against further enhanced myogenic tone and prevent further abrogated endothelium-dependent vasoconstriction in arteries during early normotensive pregnancy stage. Suggestively, although the observed changes in MMP-2 highlight its role in aortas (but not in plasma) and in utero-placental remodeling during pregnancy, further studies should investigate possible involvement of other members of the MMPs family during normotensive pregnancy stages [76]. 

Conversely, L-NAME reduced NO bioavailability and this effect was associated with increases in activity of MMP-2 in uterus, placenta and aorta, and MMP-9 in plasma and placenta of hypertensive compared to normotensive pregnant rats. Importantly, an excessive increase in MMPs activity [26,27,28] may be associated with reduced NO bioavailability [12,13,14,15,16,17,18,19], thus contributing to the pathophysiology of hypertensive pregnancy disorders, including preeclampsia. These suggestions are aligned with our previous findings showing that activation of MMPs induced by L-NAME was attenuated by doxycycline, a non-selective MMPs inhibitor, in late-hypertensive pregnant rats [77]. Accordingly, as reported in separate investigations, MMP-2 cleaved and inactivated the vasodilator calcitonin gene-related peptide (CGRP) and MMP-2 along with CGRP significantly decreased the CGRP-induced vasodilatory effect [58]. Furthermore, experiments with a highly selective gelatinase inhibitor displayed that the MMP-2 inhibition resulted in vascular relaxation in a time and concentration-dependent manner [78]. Along with these results, abnormal MMPs activation has been reported in hypertensive disorders of pregnancy, and there are studies showing that MMPs may affect the vascular function and play roles in the vascular alterations found in preeclampsia and in other hypertension-related pregnancy complications [79,80]. Moreover, under normal pregnancy, MMPs activation may be regulated by endogenous MMPs inhibitors, the TIMPs [81,82].

Endogenous NO synthesis during normotensive pregnancy is elevated in middle and late pregnancy stages but not in the early pregnancy stage [10], which support our present results. Moreover, as angiotensin II is also increased in L-NAME-treated pregnant rats at the late pregnancy stage [83] and that L-NAME enhanced mesenteric artery contractions to big endothelin-1 in normotensive pregnant rats [84], thus in our stuty, L-NAME could have caused further hyperreactivity to angiotensin II and endothelin-1 [84,85]. In addition, we previously showed that L-NAME-induced hypertension in late pregnant rats triggered the increases in MMP-2 and -9 activities in placenta and MMP-2 in uterus and these changes were associated with angiogenic imbalance which was featured by increase in soluble fms-like tyrosine kinase-1 (sFlt-1) and placental growth factor (PLGF) [77]. Moreover, we also observed in another previous investigation that increases in placental NO levels underlie the improvement in placental efficiency in late hypertensive pregnant rats [86].

Taken together, increases in endogenous NO may downregulate MMP-2 and MMP-9 during normal pregnancy, because increases in activity of these galatinases were found under reduced NO bioavailability caused by L-NAME in the present study, implying that there may be an important mechanism enrolled in hypertensive disorders of pregnancy, since reductions in NO levels as well as increases in MMP-2 and MMP-9 have been found in pregnant women complicated by gestational hypertension and preeclampsia [12,13,14,15,20,26,27,28,87]. Importantly, our results indicate that there may be a demand for increased NO with the advancement of healthy pregnancy, while reductions in NO bioavailability induced by L-NAME during middle and late pregnancy stages may result in greater activity of MMPs, and further contributes to hypertension in pregnancy. 

Moreover, endothelial dysfunction in preeclampsia may be resultant of increased oxidative stress associated with reduced NO bioavailability [88,89,90,91]. Indeed, increased oxidative stress may enhance vascular concentrations of peroxynitrite that contributes to the pathogenesis of preeclampsia [90]. There is evidence that peroxynitrite may directly activate MMPs [92], as confirmed in an experimental study showing that the endothelial function improved in the preeclamptic rat model, probably as a result of attenuated peroxynitrite formation [52]. Furthermore, it has also been proposed that angiotensin II itself may trigger the activation of MMPs. In accordance with this idea, angiotensin II induced increases in MMP-2 with concomitant decreases in TIMP-2, in a concentration-dependent manner, in human umbilical vein endothelial cells (HUVECs) isolated from umbilical cords, as previously described [93]. Also, oxidative stress generated by the enzyme nicotinamide adenine dinucleotide phosphate (NADPH) oxidase induced MMP-9 releasing, as was previously observed [94].

We also found that L-NAME increased oxidative stress in middle and late pregnancy stages, but not in early-pregnant rats compared to respective saline-treated group. Faced with these results, we may speculate that the L-NAME treatment resulted in greater responses induced by vasoconstrictors endothelin-1 and angiotensin II in middle and late stages. Giving support to this statement, recent clinical evidence demonstrated that increased angiotensin II sensitivity contributed to microvascular dysfunction in women who have had preeclampsia [95]. Also, L-NAME enhanced endothelin-1, angiotensin II, and phenylephrine-induced vasoconstrictor responses in mesenteric arterial beds and mesenteric arteries of pregnant rats and these effects were associated with increases in oxidative stress [85,96,97]. Thus, these adverse actions particularly of angiotensin II induced by L-NAME during pregnancy [85,93,97] could have induced oxidative stress in middle and late pregnancy stages [95,97]. Furthermore, increased levels of extracellular MMP inducer (EMMPRIN) were markedly elevated in women with preeclampsia as compared to levels in normotensive pregnant women [98]. The EMMPRIN is a widely expressed membrane protein of the immunoglobulin superfamily, and has been implicated in tissue remodeling and various pathological processes [24,98]. EMMPRIN stimulates the production of MMP-1, -2, -3, and -9 and may have mediated MMP regulation in endothelial cells [24,98] in the hypertensive pregnant rats of the present study. However, a future study should examine whether EMMPRIN is modulated by NO as well.

Although we have not examined antioxidant status in the present study, we may suggest that increases in oxidative stress may also contribute to MMPs activation observed in hypertensive pregnant rats. In this context, increased oxidative stress was found in middle pregnant rats treated with L-NAME, and, we may speculate that the middle pregnancy stage is more susceptible to the oxidative stress caused by L-NAME than the early pregnancy stage. Similarly, our present results also suggest that pregnancy condition is also more susceptible to these adverse effects of L-NAME compared to virgin rats, in which we found the opposite, since L-NAME-treated virgin rats displayed reduced oxidative stress compared to the respective saline-treated virgin group. Of note, although high oxidative stress was similar in normotensive and hypertensive pregnant groups at the late pregnancy stage, higher metabolic demands required for fetal development at the end of pregnancy, and the increases in oxidative stress could have been counterbalanced by abundant antioxidant defenses that occur during healthy pregnancy but not in pregnant rats treated with L-NAME [85,99,100]. Giving support to our present results, a previous study showed that there were no significant differences in lipid peroxidation in L-NAME-treated rats compared to saline-treated animals on the pregnancy day 21 [101]. Thus, we suggest that the impaired antioxidant system associated with the increase in oxidative stress caused by L-NAME [50] could have also contributed to triggering the activation of MMPs during hypertension in pregnancy [102].

Therefore, we may speculate that modification of the thin MMPs/TIMPs balance is NO-dependent and may play a key role in the uterine, placental and vascular changes associated with the physiological adaptations during pregnancy as well as MMPs/TIMPs imbalance [103] may be related to the complications of pregnancy such as hypertension in pregnancy and preeclampsia [104].

Some limitations should be considered. Firstly, redox imbalance has been reported in preeclampsia, consequently, this alteration must be further investigated, because it may also contribute to the pathogenesis of this critical condition. Secondly, elevated pro-inflammatory response has also been shown in preeclampsia [80,105,106], thus, additional studies need to address whether eNOS-derived NO may prevent upregulation of pro-inflammatory cytokines and MMPs in preeclampsia. Thirdly, although FBS was used as a positive control to normalize the inter-gel activities as previously described [46,47,48,77], careful examination of the zymograms should be evaluated in future studies using U-937 as a standard, because it is more appropriate in order to differentiate pro-enzymes and active forms of both MMPs analyzed here. Lastly, although this is the first article that shows that L-NAME-treated rats in early, middle, and late pregnancy stages exhibited reduction in circulating NO followed by increased activity of MMP-2 and MMP-9, further studies are warranted to examine the underlying mechanisms linking reduced NO and increased MMPs in hypertensive pregnancy-induced changes.

## 5. Conclusions

In summary, our data indicate that NO demand may be greater during middle and late periods of gestation, and that physiological increases in endogenous NO may negatively modulate the activity of MMP-2 and MMP-9 during a healthy pregnancy. However, reductions in NO bioavailability induced by L-NAME were associated with increases in activity of these MMPs during hypertension in pregnancy with concomitant fetal growth restriction. Therefore, our present results are in accordance with the hypothesis that NO modulates the activity of MMP-2 and MMP-9 in pregnancy and further contribute and extend earlier observations in which the main focus was the late pregnancy stage.

## Figures and Tables

**Figure 1 cells-08-01402-f001:**
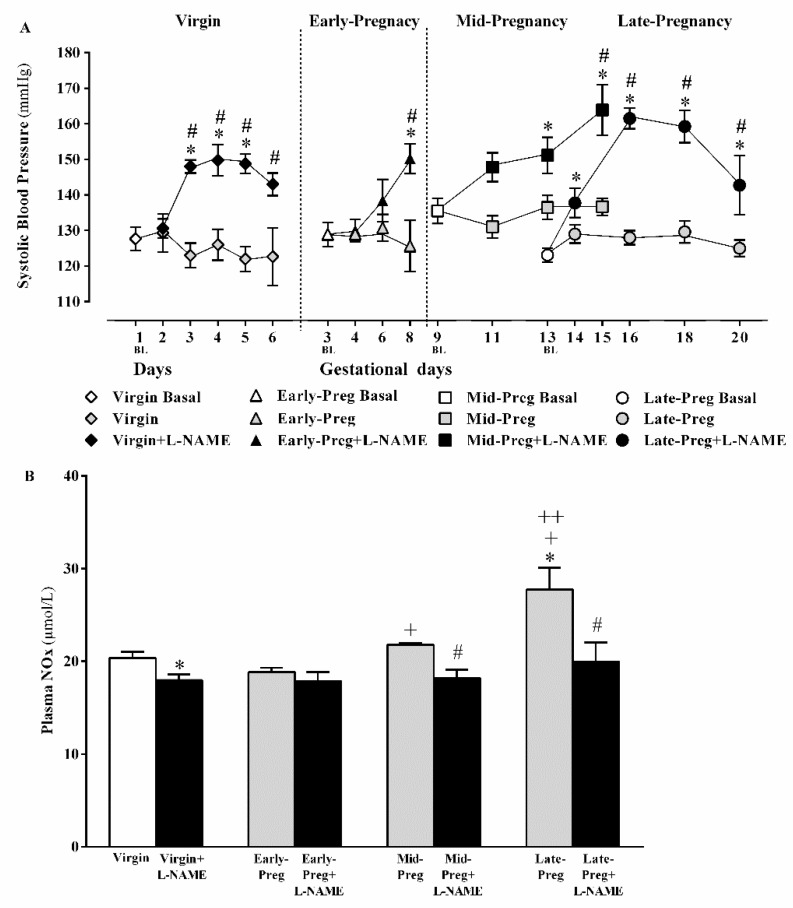
Systolic blood pressure measurements (**A**) and NO bioavailability in plasma (**B**) of virgin rats and Early, middle (Mid) and Late-pregnant rats treated with saline or L-NAME. Values are represented as mean ± SEM. In the panel A, **P* < 0.05 *versus* baseline (BL) of each group and ^#^*P* < 0.05 *versus* respective day of each group treated with saline. In the panel B, **P* < 0.05 *versus* Virgin group, ^#^*P* < 0.05 *versus* respective Mid-Preg or Late-Preg groups, ^+^*P* < 0.05 *versus* Early-Preg or Early-Preg+L-NAME groups, and ^++^*P* < 0.05 *versus* Mid-Preg or Mid-Preg+L-NAME groups.

**Figure 2 cells-08-01402-f002:**
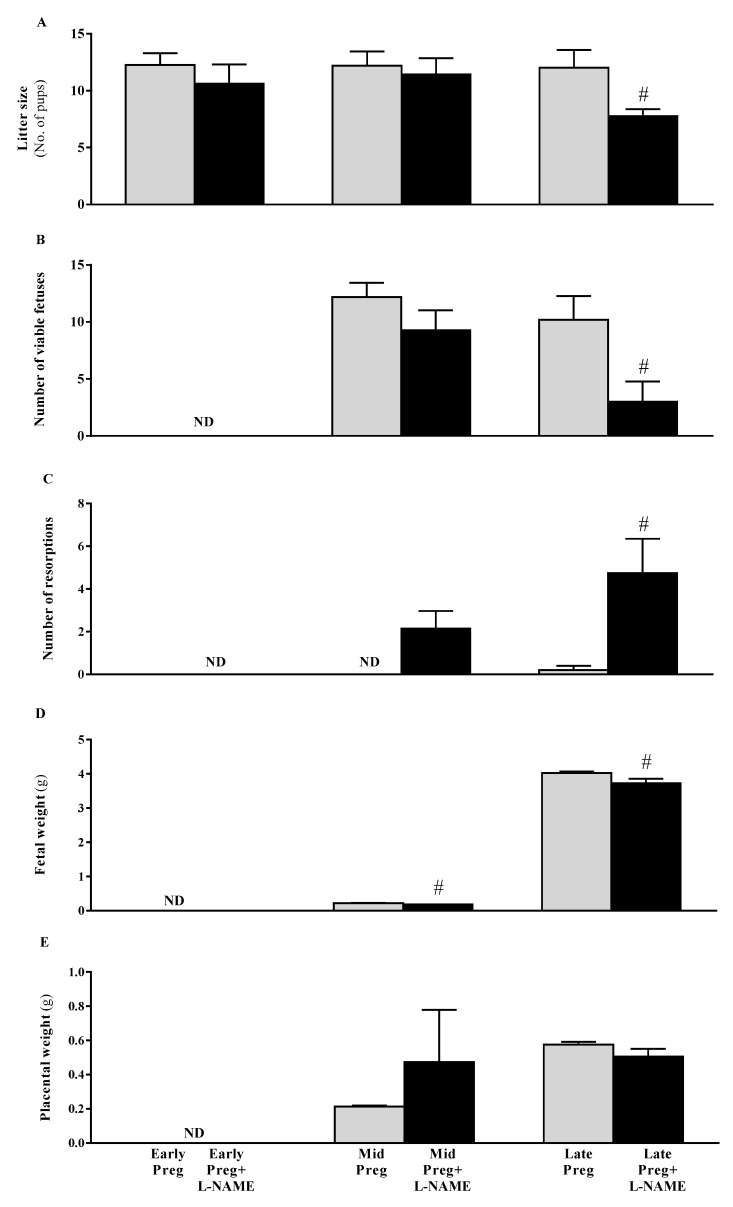
Fetal and placental parameters: (**A**) Litter size, (**B**) Number of viable fetuses, (**C**) Number of reabsorptions, (**D**) Fetal weight and (**E**) Placental weight in Early, middle (Mid) and Late-pregnant rats treated with saline or L-NAME. Values are represented as mean ± SEM. ^#^*P* < 0.05 *versus* respective saline-treated pregnant groups.

**Figure 3 cells-08-01402-f003:**
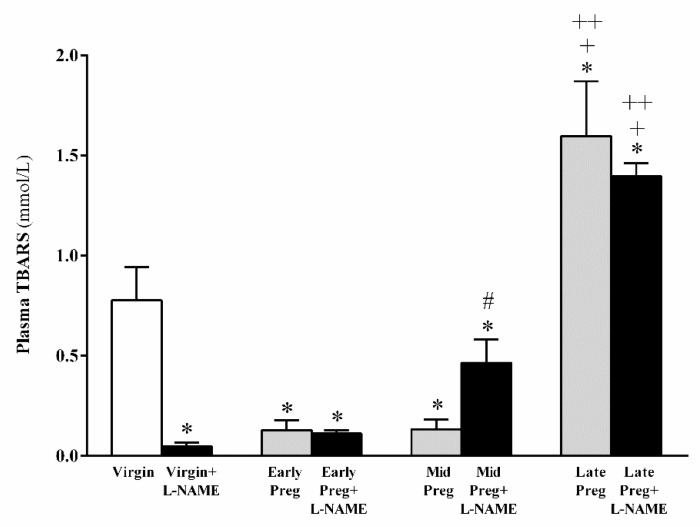
Oxidative stress of virgin rats and Early, middle (Mid) and Late-pregnant rats treated with saline or L-NAME. Values represent mean ± SEM. **P* < 0.05 *versus* Virgin group, ^#^*P* < 0.05 *versus* the Mid-Preg group, ^+^*P* < 0.05 *versus* Early-Preg and Early-Preg+L-NAME groups, and ^++^*P* < 0.05 *versus* Mid-Preg and Mid-Preg+L-NAME groups.

**Figure 4 cells-08-01402-f004:**
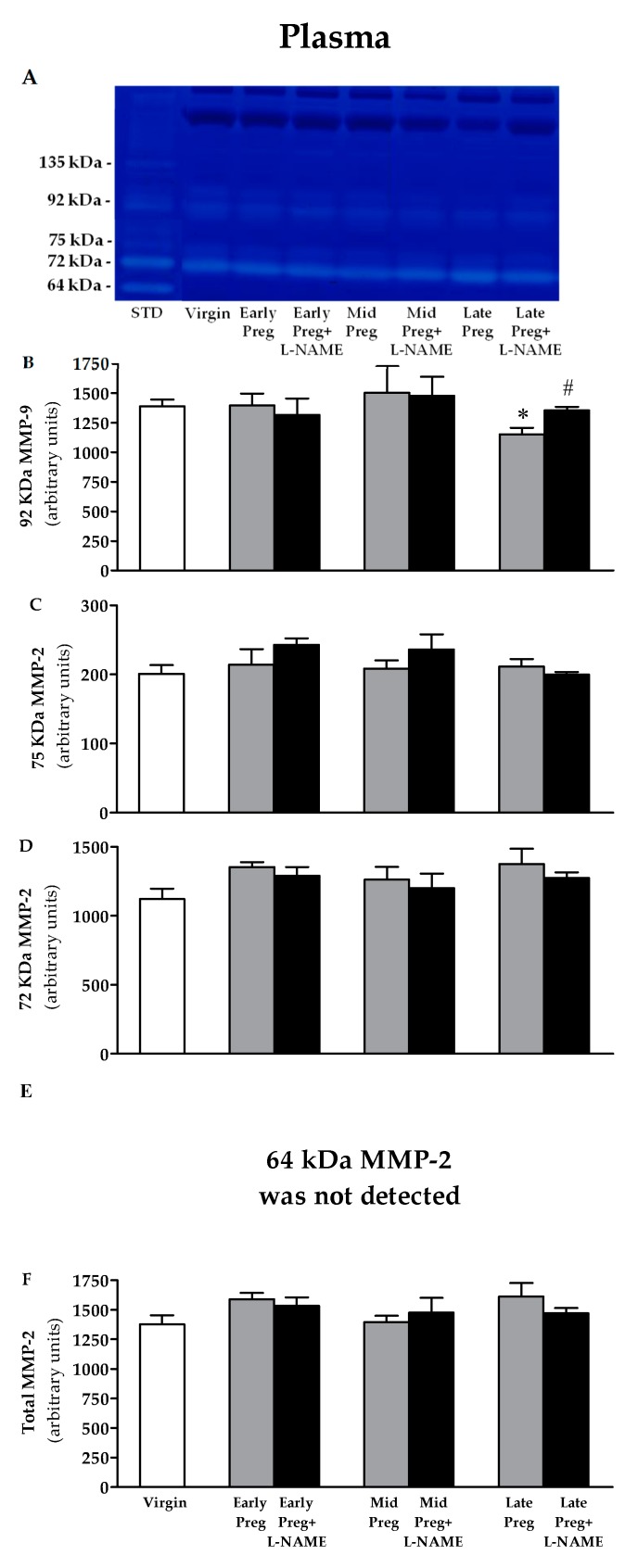
Representative zymography gel from the plasma (**A**). Gelatinolytic activity of 92 KDa MMP-9 (**B**), 75 KDa MMP-2 (**C**), 72 KDa MMP-2 (**D**), 64 KDa MMP-2 was not detected (**E**) and total MMP-2 (**F**) in virgin rats and Early, middle (Mid) and Late-pregnant rats treated with saline or L-NAME. Values represent mean ± SEM. **P* < 0.05 *versus* the Virgin group, ^#^*P* < 0.05 *versus* the Late-Preg group.

**Figure 5 cells-08-01402-f005:**
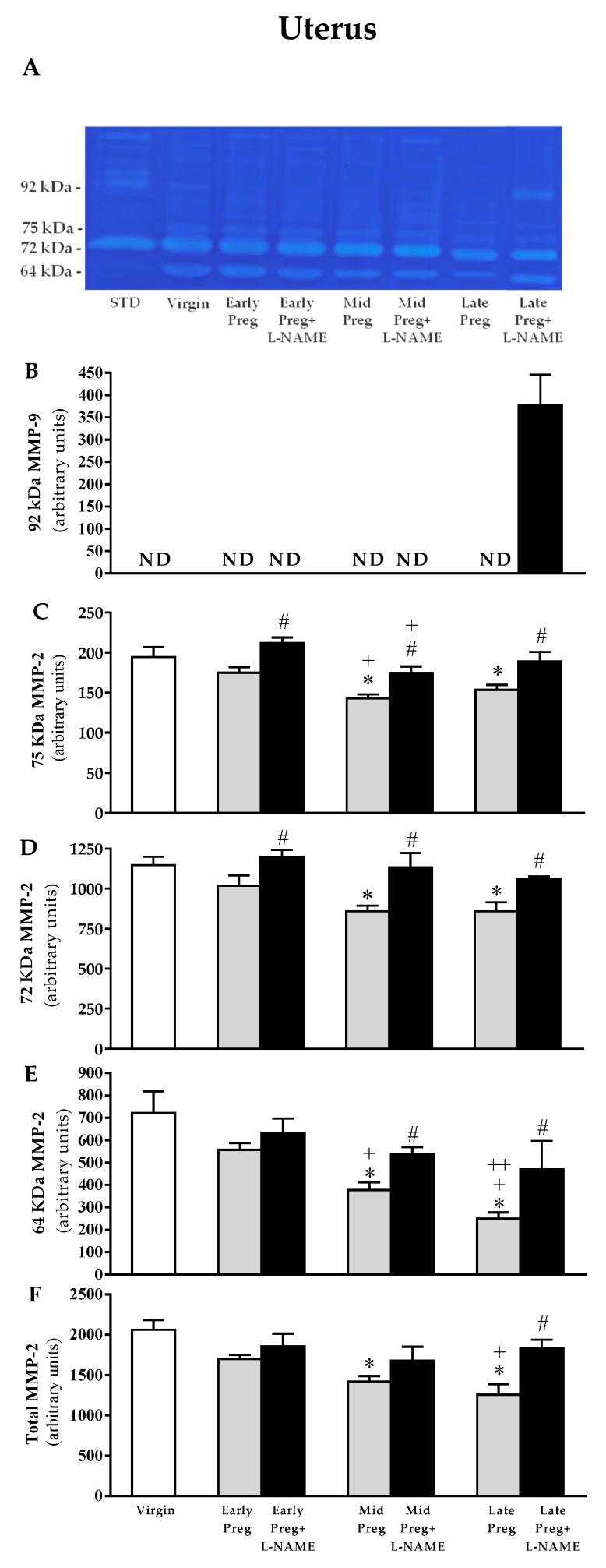
Representative zymography gel from the uterus (**A**). Gelatinolytic activity of 92 KDa MMP-9 was only detected in Late-Perg+LNAME group (**B**), and, 75 KDa MMP-2 (**C**), 72 KDa MMP-2 (**D**), 64 KDa MMP-2 (**E**), and total MMP-2 (**F**) in virgin rats and Early, middle (Mid), and Late-pregnant rats treated with saline or L-NAME. Values represent mean ± SEM. **P* < 0.05 *versus* Virgin group, ^#^*P* < 0.05 *versus* respective Early-Preg, Mid-Preg and Late-Preg groups, ^+^*P* < 0.05 *versus* Early-Preg group and ^++^*P* < 0.05 *versus* Mid-Preg or Mid-Preg+L-NAME groups.

**Figure 6 cells-08-01402-f006:**
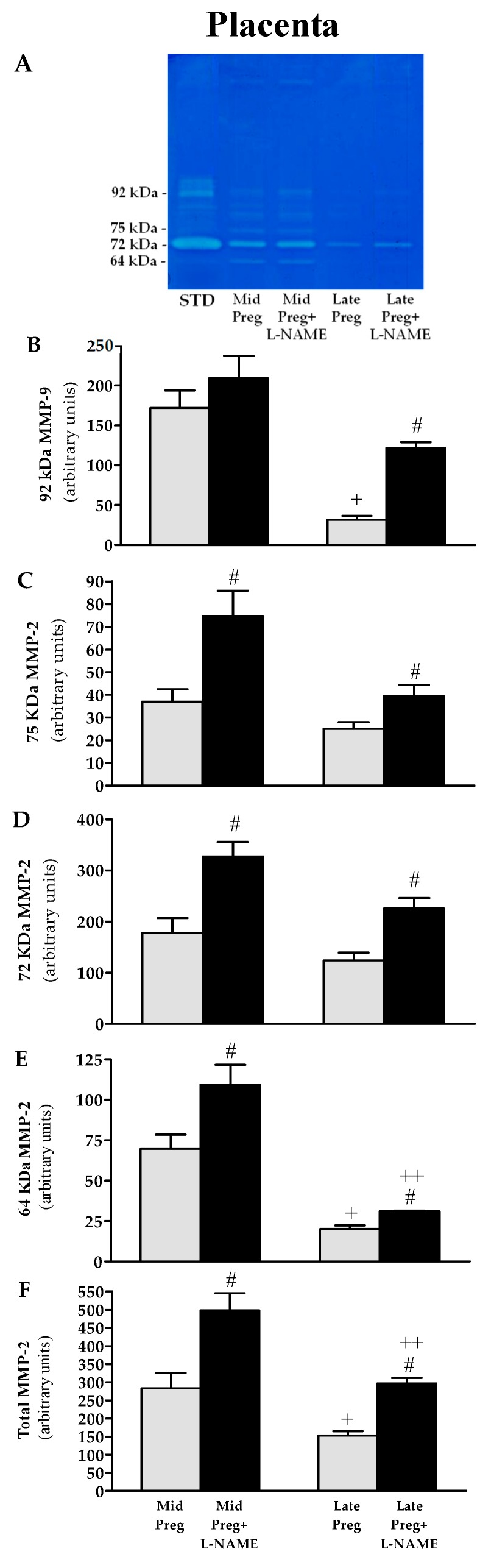
Representative zymography gel from the placenta (**A**). Gelatinolytic activity of 92 KDa MMP-9 (**B**), 75 KDa MMP-2 (**C**), 72 KDa MMP-2 (**D**), 64 KDa MMP-2 (**E**) and total MMP-2 (**F**) in middle (Mid) and Late-pregnant rats treated with saline or L-NAME. Values represent mean ± SEM. ^#^*P* < 0.05 *versus* respective saline-treated group, and ^+^*P* < 0.05 *versus* respective Mid-Preg group, ^++^*P* < 0.05 *versus* respective Mid-Preg+L-NAME group for 64 KDa and total MMP-2 activities.

**Figure 7 cells-08-01402-f007:**
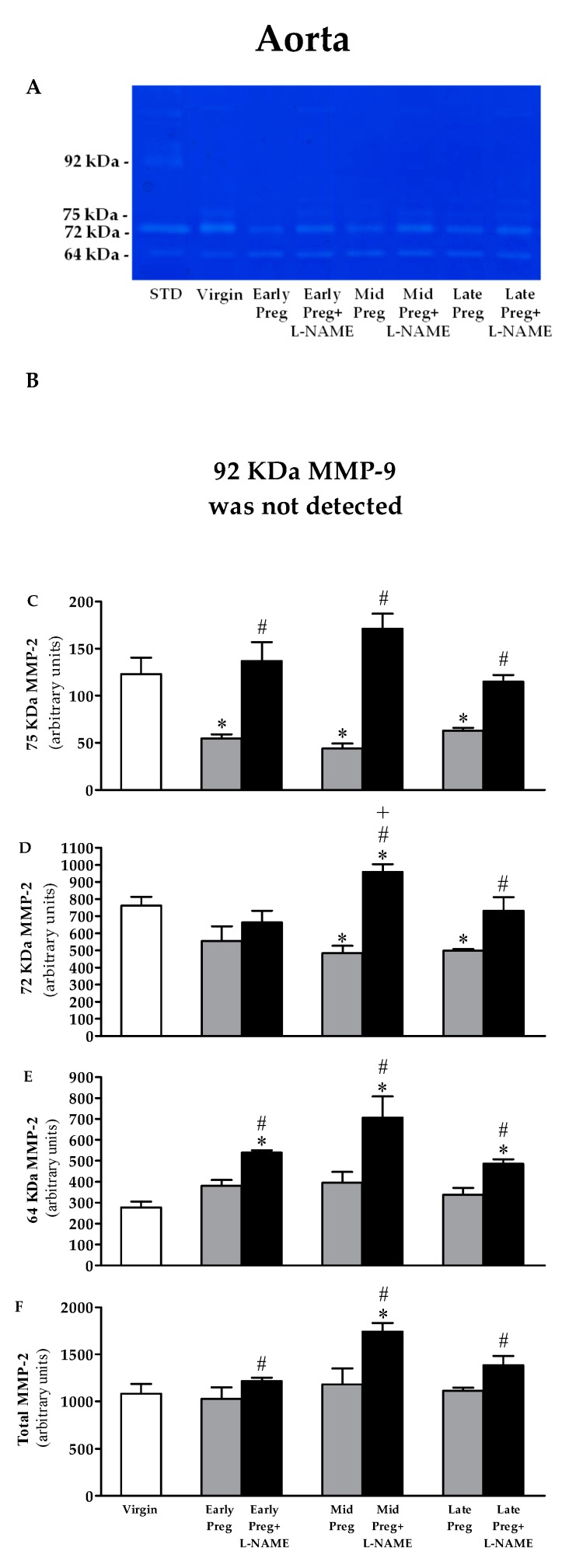
Representative zymography gel from the aorta (**A**). Gelatinolytic activity of 92 KDa MMP-9 was not detected (**B**), and, 75 KDa MMP-2 (**C**), 72 KDa MMP-2 (**D**), 64 KDa MMP-2 (**E**) and total MMP-2 (**F**) in virgin rats and Early, middle (Mid) and Late-pregnant rats treated with saline or L-NAME. Values represent mean ± SEM. **P* < 0.05 *versus* the Virgin group, ^#^*P* < 0.05 *versus* respective Early-Preg, Mid-Preg and Late-Preg groups, ^+^*P* < 0.05 *versus* the respective Early-Preg group.

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
