# Peer review of "Reductions of Circulating Nitric Oxide are Followed by Hypertension during Pregnancy and Increased Activity of Matrix Metalloproteinases-2 and -9 in Rats"

_cells, 2019, doi:10.3390/cells8111402_

Round 1

Reviewer 1 Report

The manuscript has been gratly improved, so is now suitable for publication

Reviewer 2 Report

The authors responded well to my critiques and have added additional references and detail that aids in the interpretation of their data. The paper is still fundamentally descriptive and not mechanistic, however, these data may prove useful as a foundation for future studies.  

This manuscript is a resubmission of an earlier submission. The following is a list of the peer review reports and author responses from that submission.

Round 1

Reviewer 1 Report

The manuscript by Nascimento et al aims to establish a correlation between the levels of nitric oxide and activity of matrix metalloproteinases (MMP) in plasma, placenta, uterus and aorta in the model of L-NAME-induced hypertensive pregnancy.

The established model of L-NAME treatment to induce a preeclampsia-like syndrome has been well documented in the literature. The authors show the effects of this treatment as increased blood pressure, decreased NO bioavailability in mid and late pregnancy and decreased litter's size, fetal viability and weight in late pregnancy. The authors also show the effects of L-NAME on NOx levels and markers of oxidative stress (TBARS). Variable effects were observed on MMP levels during L-NAME treatment.

One novel conclusion is that changes in NO bioavailability alter MMP expression during pregnancy.

The manuscript is well written, experiments seem well conducted and the conclusions are supported by the results.

The dose of L-NAME used (60 mg/kg/day) should be supported by references (i.e. Arch Gynecol Obstet (2005) 271:243)

In the description of zymography it should be included that the washing step (was Triton used here?) facilitates regeneration of enzymes to regain functional activity.

Do the authors have an explanation for the increased oxidative stress at mid-pregnancy in the L-NAME-treated group?

MMPs have been associated with the reduced contraction and systemic vasodilatation seen during pregnancy. However, the authors found reduced levels of MMP-2 (75 and 72kDa) in aorta and reduced MMP-9 and total MMP-2 in placenta and uterus. This need to be better explained.

Minor:

Line 168: Ealry-Preg

Line 205, first line: "has" change it to "was"

Author Response

The manuscript by Nascimento et al aims to establish a correlation between the levels of nitric oxide and activity of matrix metalloproteinases (MMP) in plasma, placenta, uterus and aorta in the model of L-NAME-induced hypertensive pregnancy.
The established model of L-NAME treatment to induce a preeclampsia-like syndrome has been well documented in the literature. The authors show the effects of this treatment as increased blood pressure, decreased NO bioavailability in mid and late pregnancy and decreased litter's size, fetal
viability and weight in late pregnancy. The authors also show the effects of L-NAME on NOx levels and markers of oxidative stress (TBARS). Variable effects were observed on MMP levels during LNAME treatment.
One novel conclusion is that changes in NO bioavailability alter MMP expression during pregnancy.
The manuscript is well written, experiments seem well conducted and the conclusions are supported by the results.
The dose of L-NAME used (60 mg/kg/day) should be supported by references (i.e. Arch Gynecol Obstet (2005) 271:243)

Response: We agree with you. Now, we have included a statement in the 2.2. Animals and experimental protocol (please see lines 98), as following:
The dose of the L-NAME (60 mg/Kg/daily) was based on previous studies [40+].

In the description of zymography it should be included that the washing step (was Triton used here?) facilitates regeneration of enzymes to regain functional activity.

Response: Yes, we have used Triton. Now, we added a statement to provide this important step in the methods (please see lines 176 and 177), as following:
...­ After electrophoresis was completed, the gels were incubated twice for 30 min at room temperature in a 2% Triton X-100 solution, washed and incubated ...

Do the authors have an explanation for the increased oxidative stress at mid-pregnancy in the L-NAME-treated group?

Response: Yes. Because angiotensin II itself increases oxidative stress [97] and there is previous evidence that supports our present results, in which increases in angiotensin II levels and expression were observed in L-NAME-treated pregnant rats [83]. Then, since increased oxidative stress was also found in middle pregnant rats treated with L-NAME in the present study, we may speculate that the middle pregnancy stage is more susceptible to the angiotensin II-induced oxidative stress caused by L-NAME than early pregnancy stages. Then, we included statements (please, see lines 694 to 699) to clarify this important observation that you raised, as following:
In this context, increased oxidative stress was found in middle pregnant rats treated with L-NAME, and, we may speculate that the middle pregnancy stage is more susceptible to the oxidative stress caused by L-NAME than early pregnancy stage. Similarly, our present results also suggest that pregnancy condition is
also more susceptible to these adverse effects of L-NAME compared to virgin rats, in which we found the opposite, since L-NAME-treated virgin rats displayed reduced oxidative stress compared to the respective saline-treated virgin group.

MMPs have been associated with the reduced contraction and systemic vasodilatation seen during pregnancy.

Response: We respectfully disagree with you on this comment. Because we and others have shown that increased MMP-2 may be associated to increased contraction and systemic vasoconstriction. Please, see lines 624 to 652, as following; Conversely, L-NAME reduced NO bioavailability and this effect was associated with increases in activity of MMP-2 in uterus, placenta and aorta, and MMP-9 in plasma and placenta of hypertensive compared to normotensive pregnant rats. Importantly, an excessive increase in MMPs activity [26-28] may be associated with reduced NO bioavailability [12-19], thus contributing to the pathophysiology of hypertensive pregnancy disorders, including preeclampsia. These suggestions are aligned with our previous findings showing that activation of MMPs induced by L-NAME was attenuated by doxycycline, a non-selective MMPs inhibitor, in late-hypertensive pregnant rats [77]. Accordingly, as reported in separate investigations, MMP-2 cleaved and inactivated the vasodilator calcitonin gene-related peptide (CGRP) and MMP-2 along with CGRP significantly decreased the CGRP-induced vasodilatory effect [58]. Furthermore, experiments with a highly selective gelatinase inhibitor displayed that the MMP-2 inhibition resulted in vascular relaxation in a time and concentration-dependent manner [78]. Along with these results, abnormal MMPs activation has been reported in hypertensive disorders of pregnancy, and there are studies showing that MMPs
may affect the vascular function and play roles in the vascular alterations found in preeclampsia and in other hypertension-related pregnancy complications [79, 80]. Moreover, under normal pregnancy, MMPs activation may be regulated by endogenous MMPs inhibitors, the TIMPs [81, 82]. Because endogenous NO synthesis during normotensive pregnancy is elevated in middle and late
pregnancy stages but not in early pregnancy stage [10], which support our present results. Moreover, as angiotensin II is also increased in L-NAME-treated pregnant rats at the late pregnancy stage [83] and that LNAME enhanced mesenteric artery contractions to big endothelin-1 in normotensive pregnant rats [84], thus in our stuty, L-NAME could have caused further hyperreactivity to angiotensin II and endothelin-1 [84, 85]. In addition, we previously showed that L-NAME-induced hypertension in late pregnant rats triggered the increases in MMP-2 and -9 activities in placenta and MMP-2 in uterus and these changes were associated with angiogenic imbalance which was featured by increase in soluble fms-like tyrosine kinase-1 (sFlt-1) and placental growth factor (PLGF) [77]. Moreover, we also observed in other previous investigation that
increases in placental NO levels underlie the improvement in placental efficiency in late hypertensive pregnant rats [86].

However, the authors found reduced levels of MMP-2 (75 and 72kDa) in aorta and reduced MMP-9 and total MMP-2 in placenta and uterus. This need to be better explained.

Response: We agree with you that the present results in which reduced MMP-2 (75 and 72kDa) in aorta and reduced MMP-9 and MMP-2 in placenta and reduced MMP-2 in uterus and that MMP-9 was only detected in Late-Preg+L-NAME group of the uterus deserve an explanation to clarify our findings to the readers. So, we now included a statement and a paragraph in the discussion (please see lines 597 to 623), as following; In general, we observed that during healthy pregnancy, there are increases in NO bioavailability followed by decreases in the MMP-2 activity in uterus, placenta and aorta and decreases in the MMP-9 activity in plasma and placenta in normotensive pregnant rats. Although some isoforms presented no changes in their activities in normotensive pregnant compared to virgin rats, our present results suggest that the reduced activity of MMPs in healthy pregnant rats may be related to the MMPs regulation mechanisms in middle and late pregnancy stages. It is possible that the aorta, uterus and placenta as potential sources of MMPs may have finite capacity for gelatinolytic activity and that may be downregulated by the increases in
endogenous NO, as demonstrated previously in vitro [29], and our data suggest that downregulation of MMPs activity occurs particularly during the middle and late pregnancy stages. According to this idea, our results also showed no significant changes neither in activity of MMPs (in plasma, utero and placenta) nor in circulating NO at the early pregnancy stage, implicating a key role played by NO in the downregulation of these gelatinases in the fetal-placental circulation during late pregnancy stages. Also, most of the arterial,
uterine and placental remodeling occurs during the peri-implantation period and during fetal and organ development in early and middle gestation and further arterial, uterine and placental remodeling may not be needed during late pregnancy stage. Interestingly, we unexpectedly also detected decreases in activity of MMP-2 (75KDa) in aorta at the early pregnancy stage compared to Virgin rats, suggesting that this downregulation mechanism may be associated with early pregnancy-induced changes, but it is not related to NO signaling, i.e., during middle and late pregnancy stages endogenous NO synthesis may downregulate MMPs, while during early pregnancy stage this downregulating mechanism may not be dependent of endogenous NO synthesis, but this finding implicates that early pregnancy-induced adaptations protect against further enhanced myogenic tone and prevent further abrogated endothelium-dependent
vasoconstriction in arteries during early normotensive pregnancy stage. Suggestively, although the observed changes in MMP-2 highlight its role in aortas (but not in plasma) and in utero-placental remodeling during pregnancy, further studies should investigate possible involvement of other members of the MMPs family during normotensive pregnancy stages [76].

Minor:
Line 168: Ealry-Preg
Line 205, first line: "has" change it to "was"

Response: We thank you for your careful revision, and, we now corrected them as your observations. Please see revised lines 213 and 240

Reviewer 2 Report

In this manuscript, the authors study if MMP's activation is regulated by NO bioavailability during pregnancy in an animal model. Using this experimental model with L-NAME, hypertension is induced in the different experimental groups, showing reduced NOx concentrations in plasma.

a) The main findings are focused in MMP-2 and MMP-9 activity but I consider that the results are not supported by the zymography images shown. 

1. The use of FBS as standard for MMP-2 and -9 presence and activity is not usual. We can observe in the zymography gels several bands but, how can the authors define the molecular weight of each one? The use of U-937 supernatant is more appropriate to differentiate between the pro-enzyme and active forms of both MMPs. This is critical to get truthful results.

2. According to literature, the standard methodology for detection of MMP-2 includes 72 kDa and 64 kDa forms. Why are the authors including a 75 kDa form in their analysis? 
 3. Additionally, zymography gel for the plasma samples (Fig 4A) does not show the 64 kDa form in the standard,  therefore it is important to select a more representative image for this experiment. 

4. Zymography gel for the uterus samples (Fig 5A) show bands of approximately 92 kDa corresponding to MMP-9 in the late preg+ L-NAME. Although this result may not be statistically significant it should be included. Importantly the 92 kDa band is not showed in the standard lane.

5. Overall, zymography gels show different lysis bands that were not consider in the results and discussion sections.

b) The methodology section is not structured in a clear manner and should include a description of the study design as well as a more detailed description of the study groups.

1. The study design description should be presented in the top of the section

2. The description of the study groups must be restructured. It is not well justified, neither included in the aims of the study why are the authors using the analysis on the groups of virgin rats with without induced hypertension.

c)English editing and style must be further revised, and a syntax correction in several paragraphs is needed.

English grammar is incorrect in several lines, for example, line 11- Hypertensive pregnancy HAVE been associated....

Several paragraphs must be rephrased in order to deliver a correct idea, for example, the results section in the abstract (Lines 17-24)  should be fully restructured, due to a lack of coherence, it is nearly impossible for the reader to understand the main findings of the study in one of the most important areas of the manuscript.

Paragraph in lines 40-42 does not make any sense.

The aim section (Lines 56-59) is not well constructed because is just a description of what was done.

d) All of the figure legends must be revised and rephrased. The titles used in the results section must be simplified and some of them should be presented in a separated section if required. The title of the manuscript is overstated and must mention that the study was performed in an animal model, even in the abstract

e) Discussion is not consistent with the results and some of the findings are not exploited in this section.

Author Response

a) The main findings are focused in MMP-2 and MMP-9 activity but I consider that the results are not supported by the zymography images shown.

We thank you for your careful revision and we agree with you that our representative zymography gels are not supporting our findings. Then, we reviewed all zymography images and we replaced them by the original color pictures of each gel with areas corresponding to MMP gelatinolytic activity appearing as clear bands against a blue background. Please, see our color gel images in the revised version of the manuscript.

1. The use of FBS as standard for MMP-2 and -9 presence and activity is not usual. We can observe in the zymography gels several bands but, how can the authors define the molecular weight of each one? The use of U-937 supernatant is more appropriate to differentiate between the pro-enzyme and active forms of both MMPs. This is critical to get truthful results.

Response: We agree with you that the use of U-937 supernatant is more appropriate to differentiate pro- and active forms of the MMPs. However, we and others have used FBS as positive control to normalize the inter-gel activities of these MMPs, as previously described [Vascul Pharmacol. 2019, 116:36-44, 77]. Even so, we now included a statement that careful examination of the zymograms should be evaluated in a future study using U-937 as standard. Please, see this sentence in the limitations, lines 719 to 722, as following:

Thirdly, although FBS was used as positive control to normalize the inter-gel activities as previously described [46-48, 77], careful examination of the zymograms should be evaluated in future studies using U- 937 as standard, because it is more appropriated to differentiate pro-enzymes and active forms of both MMPs analyzed here.

2. According to literature, the standard methodology for detection of MMP-2 includes 72 kDa and 64 kDa forms. Why are the authors including a 75 kDa form in their analysis?

Response: Although 72 kDa and 64 kDa MMP-2 bands are the most human disease-related, previous studies have found the 75 kDa band in rodents [Vascul Pharmacol. 2019, 116:36-44, Free Radic Biol Med. 2019, 130:234-243, 77], in which similar increases in 72 kDa MMP-2 were found under the same methods used in the present study. Moreover, 75 kDa band may be correspondent to a modified form of the MMP-2 or a product of the MMP-9 activation resultant of the action promoted by neutrophil elastase as previously suggested [Circulation. 2000, 18;101(15):1833-9; FEBS Letters. 1997, 402:111-115].

3. Additionally, zymography gel for the plasma samples (Fig 4A) does not show the 64 kDa form in the standard, therefore it is important to select a more representative image for this experiment.

Response: We apologize for this oversight and we thank you again for your careful revision. Then, we now selected a more representative zymography gel for the plasma in which 64 kDa form appears in the standard. Please, see our Figure 4A in the revised version of the manuscript.

4. Zymography gel for the uterus samples (Fig 5A) show bands of approximately 92 kDa corresponding to MMP-9 in the late preg+ L-NAME. Although this result may not be statistically significant it should be included. Importantly the 92 kDa band is not showed in the standard lane.

Response: We thank you again for your careful examination, and we apologize for this oversight. Now, we selected a more representative zymography gel (Figure 5A) that shows 92 kDa band in the standard lane and we included this result (please, see Lines 301 and 302) of the zymogram analysis for Mid-Preg and Late-Preg+L-NAME groups, although these results were not statistically significant (Figure 5B).

5. Overall, zymography gels show different lysis bands that were not consider in the results and discussion sections.

Response: We expected to find other bands because a recent study showed additional high molecular weight bands corresponding to ~200 kDa and ~135 kDa in gelatin zymography of homogenates of placental, uterus and artery of Late pregnant rats [76]. Authors revealed bands ~200 kDa and ~135 kDa clearly discernible at 2 μg protein concentration and they performed gelatin zymography using 2 μg protein for loading as previously reported [76]. Then, authors stated that careful examination of the zymograms revealed that these two additional bands at ~200 kDa and ~135 kDa were more visible in the uterus compared to placenta and uterine artery, and analysis of the bands intensity showed that those two additional bands were reduced in the placenta, uterus and uterine artery of hypertensive pregnant rats compared to normotensive pregnant rats [76]. Face with these previous findings, we have performed gelatin zymography using 2 μg, 5 μg and 10 μg protein for loading. However, after careful examination, we observed no additional high molecular weight
band corresponding to ~135 kDa in gelatin zymography of homogenates of placental, uterus and uterine arteries, except in FBS that was used as positive control to normalize the inter-gel activities of these MMPs in plasma. Please, see our representative zymography gel from the plasma (Figure 4A, page 9).

b) The methodology section is not structured in a clear manner and should include a description of the study design as well as a more detailed description of the study groups.
1. The study design description should be presented in the top of the section

Response: We agree with you and in order to clarity to the readers, we now presented the experimental design in the top of this section. Please, see lines 77 to 102, as following:
The experimental groups consisted of age-matched virgin and pregnant rats that were randomly divided in 8 groups (n = 8 animals per group) as following:
1 - Normotensive virgin rats (Virgin group) received daily intraperitoneal (i.p.) injections of saline for 6 consecutive days;
2 - Hypertensive virgin rats (Virgin+L-NAME group) received daily i.p. injections of L-NAME for 5 consecutive days;
Pregnant rats were studied according to the different pregnancy stages, which correspond to the three gestational trimesters:
3 - Normotensive early pregnancy (Early-Preg group): pregnant rats received daily i.p. injections of saline from pregnancy day 3 to 8;
4 - Hypertensive early pregnancy (Early-Preg+L-NAME group): pregnant rats received daily i.p. injections of L-NAME from pregnancy day 3 to 8;
5 - Normotensive middle pregnancy (Mid-Preg group): pregnant rats received daily i.p. injections of saline from pregnancy day 9 to 15;
6 - Hypertensive middle pregnancy (Mid-Preg+L-NAME group): pregnant rats received daily i.p. injections of L-NAME from pregnancy day 9 to 15;
7 - Normotensive late pregnancy (Late-Preg group): pregnant rats received daily i.p. injections of saline from pregnancy day 13 to 20;
8 - Hypertensive late pregnancy (Late-Preg+L-NAME group): pregnant rats received daily i.p. injections of L-NAME from pregnancy day 13 to 20.
The dose of the L-NAME (60 mg/Kg/daily) was based on previous studies [40]. Moreover, experimental design was based on previous studies [35, 36] and we included virgin rats in order to investigate whether LNAME could have the same capacity to induce hypertension in virgin rats as well as whether hormonal
changes during pregnancy stages could affect significantly the effects of L-NAME at the different pregnancy stages.

2. The description of the study groups must be restructured. It is not well justified, neither included in the aims of the study why are the authors using the analysis on the groups of virgin rats with without induced hypertension.

Response: We agree with you and in order to clarity to the readers why we used normotensive and hypertensive virgin rats in the present study, we now included a statement in the lines 99 to 102, as following:
¡­ and we included virgin rats in order to investigate whether L-NAME could have the same capacity to induce hypertension in virgin rats as well as whether hormonal changes during pregnancy stages could affect significantly the
effects of L-NAME at the different pregnancy stages.

c) English editing and style must be further revised, and a syntax correction in several paragraphs is needed.

Response: We thank you again for your careful revision. Then, we revised whole manuscript and asked a native speaker for correcting the syntax in several paragraphs, and we highlighted all of them in red in the revised version of the manuscript.

English grammar is incorrect in several lines, for example, line 11- Hypertensive pregnancy HAVE been associated....

Response: We apologize for this oversight. Please, see our revised sentence in line 11.

Several paragraphs must be rephrased in order to deliver a correct idea, for example, the results section in the abstract (Lines 17-24) should be fully restructured, due to a lack of coherence, it is nearly impossible for the reader to understand the main findings of the study in one of the most important areas of the manuscript.

Response: We agree with you that several paragraphs and main findings of the study were unclear and was not spontaneous for the readers to understand the study. Then, we now rephrased these sentences. Please, see our revised version for the abstract highlighted in red.

Paragraph in lines 40-42 does not make any sense.

Response: We totally agree with you. Now, we revised this paragraph to clarify to the reader what is the main idea that we want to share. Please, see our revised paragraph in the lines 39 to 43, as following:
There are increases in endogenous NO synthesis in normotensive pregnancy [5, 6, 10, 11], while hypertensive disorders of pregnancy are associated with reductions of NO bioavailability in humans [12-16] and in animals [17-19]. Since healthy gestations undergo significant utero-placental and vascular adaptations [7, 20] and that MMPs may also play important roles in tissue remodeling [21], ¡­..

The aim section (Lines 56-59) is not well constructed because is just a description of what was done.

Response: We agree with you. Now, we revised the last paragraph in the Introduction in order to provide to the readers the aim of the present study. Please, see this rephrased paragraph in the lines 58 to 64, as following:
Therefore, this study aimed to examine whether decreases in circulating NO may be associated with changes in activities of MMP-2 and MMP-9 in early, middle and late stages of hypertension in pregnancy. To this end, we investigated the changes in systolic blood pressure, NO bioavailability and oxidative stress in plasma as well as gelatinolytic activities of MMP-2 and MMP-9 in plasma and in important tissues that undergo extensive remodeling and physiological adaptations during pregnancy, including uterus, placenta
and abdominal aorta in specific stages of normotensive pregnancy and L-NAME-induced hypertensive pregnancy in rats.

d) All of the figure legends must be revised and rephrased. The titles used in the results section must be simplified and some of them should be presented in a separated section if required. The title of the manuscript is overstated and must mention that the study was performed in an animal model, even in the abstract

Response: We totally agree with you. Now, we changed significantly all figure legends and the titles used in the Results section. Also, we agree with you that first version for the title of the manuscript was overstated. So, we also mentioned that this study was performed in rats in abstract and in the
title of the manuscript as well. Please, see our revised version regarding to these points that we now highlighted in red in the revised version of the manuscript.

e) Discussion is not consistent with the results and some of the findings are not exploited in this section.

Response: We agree with you that our results deserve improved explanations to clarify to the reader our findings. So, we now revised whole discussion, included paragraphs and discussed some of the findings that were not previously explored. Please, see the new paragraphs that we highlighted in red in the Discussion.

Reviewer 3 Report

This study examines the effects of NOS inhibition on rat pregnancy.  Different gestational ages were considered, and the principal findings are that loss of NO signaling resulted in systolic hypertension and reduced fetal but not placental weights. It was also associated with an increased incidence of resorptions, which decreased litter size.  These maternal and fetal effects have been shown in earlier studies of NOS inhibition, as this is a well-established animal model for human preeclampsia, and are generally known to worsen with gestational age, since nutritional demands increase as fetuses grow and mid- to late-pregnancy interventions exhibit more serious effects.  While some of the results complement and extend earlier observations, their offering is more along the lines of additional detail rather than significant new knowledge.

The findings regarding the matrix effects of NOS inhibition are more novel and indicate that increased bioavailability of NO in later pregnancy is associated with decreases in MMP-2 and MMP-9 activity in several tissues; accordingly, this effect is absent or reversed in animals undergoing NOS inhibition.  The prooxidant effects of NOS inhibition appears to also contribute to these effects later in pregnancy.

Although the MMP findings are of interest, as changes in these compounds have been noted in preeclamptic women (as has reduced NO bioavailability), Several important questions are not answered or addressed, and this limited my overall enthusiasm for the scientific merit of this manuscript. 

First, the experimental evidence regarding NO availability and/or oxidative stress affecting MMP activity is indirect, and there is no investigation of underlying mechanisms. 

Second, systemic NOS inhibition from L-NAME injections affects all of the NOS isoforms, and it is not clear whether these changes in matrix events in different tissues (plasma, placenta, uterine and aortic tissues were examined) are due to loss of endothelial influence (eNOS), neural function (nNOS) or other, e.g. immune events associated with the inducible (iNOS) form of this enzyme. Because of this, understanding the primary specific mechanism is difficult.  

Third, the linkage between NOS inhibition/hypertension and altered MMP activity is not explicit.  MMPs are involved in vascular remodeling, and may alter vasoactive influences, but how these lead to hypertension is not clear. Resistance arteries were not studied, and placental effects may lead to a host of other consequences, e.g. increased sFlt-1 production, etc.   If there are other studies in this regard that point to a particular mechanism linking altered MMP activity with vascular or placental dysfunction, they should be cited and discussed.

Fourth, it is not clear whether the MMP effects are a direct consequence of altered NO bioavailability/signaling, or are secondary to altered PKG signaling, for example?  How does a change in MMP-2 and MMP-9 activity lead to hypertension or fetal loss? Does hypertension itself (increased intravascular pressure) contribute to their expression?

The subject is intriguing, as are some of the findings, in view of earlier studies that have shown decreased NO availability in preeclamptic women, as well as altered MMP (and particularly MMP-2 and MMMP-9) activity, but the absence of a contextual or mechanistic basis makes this study associative and descriptive. Regrettably, and as stated above, some of the findings (e.g. that NOS inhibition induces hypertension during pregnancy, and that this results in fetal resorption and reduced growth) are well-established. Hence, their confirmatory value is low.

Finally, the NOS inhibited rat has been widely used in earlier studies as a model for human preeclampsia for more than 20 years, and one would not know this from the Introduction or Discussion, even though the number of referenced paper is high (>50).  It would be helpful if the authors  discussed the relevance of this model to human preeclampsia, since a number of studies that have shown reduced NO bioavailability in preeclamptic women, along with changes in MMP activity. This would make the findings more relevant to human disease.  

Author Response

Please see the attached response.
